# Changes in Blood Metabolites and Immune Cells in Holstein and Jersey Dairy Cows by Heat Stress

**DOI:** 10.3390/ani11040974

**Published:** 2021-03-31

**Authors:** Sang Seok Joo, Sang Jin Lee, Da Som Park, Dong Hyeon Kim, Bon-Hee Gu, Yei Ju Park, Chae Yun Rim, Myunghoo Kim, Eun Tae Kim

**Affiliations:** 1Department of Animal Science, College of Natural Resources & Life Science, Pusan National University, Miryang 50463, Korea; ssjoo7680@gmail.com (S.S.J.); wjrm4862@naver.com (S.J.L.); parkdasome@gmail.com (D.S.P.); zx7777777@pusan.ac.kr (Y.J.P.); yeoul00h@naver.com (C.Y.R.); 2Dairy Science Division, National Institute of Animal Science, Rural Development Administration, Cheonan 31000, Korea; kimdh3465@korea.kr; 3Life and Industry Convergence Research Institute, Pusan National University, Miryang 50463, Korea; g.bonhee@gmail.com

**Keywords:** heat stress, immune response, PBMC, immune physiological changes, metabolism

## Abstract

**Simple Summary:**

As global temperatures rise, thermal stress can be a major problem affecting cows. If they are subjected to heat stress, they are likely to exhibit abnormal metabolic reactions and affect their immune system. However, the relationship between metabolism and immunity during thermal stress and these crosstalk mechanisms remain unclear. Therefore, the aim of this study was to understand the changes in blood immune cell response with the physiological metabolism changes of Holstein and Jersey cows through the biochemistry and flow cytometry branches under thermal stress conditions. We found that various blood metabolites were reduced in both Holsteins and Jerseys by heat stress conditions. There were breed-specific variations in the immune cell population in Holstein and Jersey cows under different environmental conditions. The main findings of this study provide information on the metabolism and immunity changes of two types of cow under heat stress, broadening the potential relationship of these changes.

**Abstract:**

Owing to increasing global temperatures, heat stress is a major problem affecting dairy cows, and abnormal metabolic responses during heat stress likely influence dairy cow immunity. However, the mechanism of this crosstalk between metabolism and immunity during heat stress remains unclear. We used two representative dairy cow breeds, Holstein and Jersey, with distinct heat-resistance characteristics. To understand metabolic and immune responses to seasonal changes, normal environmental and high-heat environmental conditions, we assessed blood metabolites and immune cell populations. In biochemistry analysis from sera, we found that variety blood metabolites were decreased in both Holstein and Jersey cows by heat stress. We assessed changes in immune cell populations in peripheral blood mononuclear cells (PBMCs) using flow cytometry. There were breed-specific differences in immune-cell population changes. Heat stress only increased the proportion of B cells (CD4–CD21+) and heat stress tended to decrease the proportion of monocytes (CD11b+CD172a+) in Holstein cows. Our findings expand the understanding of the common and specific changes in metabolism and immune response of two dairy cow breeds under heat stress conditions.

## 1. Introduction

Global warming has recently accelerated, causing adverse effects in agricultural areas, including the livestock industry. High temperature and humidity are detrimental to the productivity of domestic animals [1]. Heat stress that derives from hot and humid environments has negative effects on livestock performance, such as decreased body weight, average daily gain, and growth rate [2]. Among the various livestock species, dairy cows are extremely sensitive to a high-temperature and high-humidity environment, which has a significant impact on productivity, especially milk production [3].

Many studies have reported that heat stress causes a decrease in dry matter intake (DMI) and milk yield compared with moderate-temperature conditions [4]. The decreased DMI in high-temperature weather precedes the reduction in milk production. The endocrine and metabolic systems of dairy cows have many important interactions during their lives [5]. For example, the reduced nutrient intake during high-temperature stress conditions causes abnormal hormonal and metabolic changes in dairy cows [6]. In recent studies, dairy cows exposed to a temperature and humidity challenge in controlled-climate chambers displayed the characteristic heat stress responses to short-term heat exposure [7]. The dairy cows were shown to reduce their feed intake and milk yield, in addition to initiating metabolic and endocrine adaptations through the alteration of insulin and prolactin. Another study focused on the liver metabolism of glucose in heat-stressed dairy cows [8]. The concentrations of blood amino acids were decreased in heat-stressed dairy cows, indicating the increased conversion of metabolites into glucose to stabilize glucose homeostasis during hot and humid weather.

High temperature and humidity also affect the immunity of dairy cows. It has been reported that heat stress influences various immune cells in heat-stressed dairy cows [9]. In neonatal dairy cows, acute brief heat stress alters the total white blood cell and immune function in calves [10]. Heat stress changes the innate immune function by altering cellular interactions with pathogens and changing acute-phase cytokines and pathogen recognition molecules that alter immune responses. In another study, the leukocyte population was affected under heat stress conditions in dairy cows [11]. The leukocyte population was decreased in both growing and adult dairy cows during summer compared with during other seasons.

In addition, it is widely known that the two representative dairy cow breeds, Holstein and Jersey, have different capacities to manage heat stress in their own way. Several researchers have suggested that Jersey cows are more tolerant than Holstein cows in terms of milk yield [12,13,14]. The decline in milk yield by Holstein cows was more rapid than that by Jersey cows across a range of temperature–humidity indices (THI) from 72 to 84 [15]. However, Jersey cows tended to have cooler rectal temperatures across the same range of THI, with Holsteins having a 0.3 °C greater body temperature than Jerseys [16].

Recently, many dairy cow studies have been conducted to increase our understanding of the mechanism of heat stress responses in nutritional and immunological aspects. However, a harmonious study of these two aspects is lacking. Additionally, there is very limited information available regarding heat tolerance and its effect on these two representative dairy cow breeds.

Therefore, the objective of this study was to understand the dynamics of the physiological changes and immune responses of Holstein and Jersey dairy cows under heat stress conditions.

## 2. Materials and Methods

### 2.1. Experimental Animals

All experimental procedures, including dairy cow care and blood collection for the animal experiments, were performed in accordance with guidelines from the Institutional Animal Care and Use Committee of the National Institute of Animal Science (NIAS), Rural Development Administration, Republic of Korea (Approval Number: NIAS-2019107). From the NIAS cow herd, 8 Holstein cows (52 ± 7.9 months old; parity numbers 2.4 ± 0.5) and 8 Jersey cows (57 ± 9.0 months old; parity numbers 2.4 ± 0.5) were used. The average body weight and milk yields of the Holstein cows were 665 ± 26.4 kg and 38.5 ± 4.5 kg/day in May and 655 ± 32.9 kg and 30.8 ± 5.3 kg/day in August, respectively. The average body weight and milk yields of the Jersey cows were 484 ± 23.6 kg and 30.4 ± 2.6 kg/day in May and 475 ± 16.0 kg and 24.2 ± 3.2 kg/day in August, respectively. The mean milk somatic cell count (SCC) was 31,510 ± 19,125 cells/mL for Holstein cows and did not exceed 178,000 cells/mL in the maximum, and 63,293 ± 8253 cells/mL for Jersey cows and did not exceed a maximum of 174,000 cells/mL. During the experimental period, SCC was maintained at less than 200,000 cells/mL without infection or physiological stress [17]. The Holstein and Jersey dairy cows used in this study had similar days in milk (DIM). In May, DIM of Holstein and Jersey cows were 139 ± 8.0 and 128 ± 15.0, and the DIM of Holstein and Jersey cows were 234 ± 8.0 and 223 ± 15.0 in August, respectively.

All experimental cows were housed in an individual tie stalls in an enclosed barn (Cheonan, Korea) and fed the same diet twice daily. The barn was equipped with a fan ventilation system, which was turned on at 09:00 h and turned off at 18:00 h. The cows’ diet was formulated to meet their energy requirements, and to preclude the selection of dietary components, they were fed total mixed rations (TMR). A total of 20 kg of feed was individually offered at 09:00 h and 16:00 h daily for each cow on a limited basis, which reduced the impact on feed intake. There were no orts from any of the cows. All cows had free access to a water source shared among stalls. The ingredients and nutrients of the diet are shown in Table 1. The digestible energy (DE) and metabolizable energy (ME) levels in the feed offered exceeded the requirements of Korean Feeding Standard for dairy cattle (2017) [18] for mid-lactation cows (milk yield 30 kg, milk fat 4.3%, milk protein 3.4%, ME 52.0 Mcal/day).

### 2.2. Measurement of Temperature–Humidity Index, Respiration, and Rectal Temperature

To determine the effects of heat stress on dairy cows induced by seasonal change, two distinct annual periods with normal and heat stress environmental conditions, based on the temperature–humidity index (THI), were used. A Testo 174H-Mini temperature and humidity data logger (Testo Korea Ltd., Seoul, Korea) was used to monitor temperature and humidity at 14:00 h for a week. The THI equation was devised by the National Research Council in 1971 [19]. The average monthly temperature and relative humidity in April and May together, and July and August were calculated using the Korea Meteorological Administration (KMD) records. The environmental conditions were set according to the THI value (THI < 71: comfort; THI < 72 to < 79: mild heat stress; THI < 80 to < 89: moderate heat stress; THI > 90 severe heat stress) [20]. The comfort range was regarded as the “normal environmental condition” and the moderated heat stress was regarded as the “high-heat environmental condition”. The dairy cows were exposed to different environmental conditions for at least 7 consecutive days based on THI. Respiration rates and rectal temperature were examined by following established protocols [21]. 

### 2.3. Blood Collection and Sample Preparation

After measuring THI and rectal temperature, the dairy cows were fixed for blood sampling. The blood samples from the jugular vein were immediately collected only once in normal environmental conditions (May 3rd, THI = 69.6, 24.4 °C, and 36.3% relative humidity (RH)) and high-heat environmental condition (August 9th, THI = 87.5, 35.5 °C, and 59.6% RH), and then put into two types of Vacutainer tube (BD Vacutainer, Becton Dickinson Co., Franklin Lakes, NJ, USA)—plastic whole-blood tubes, spray-coated with silica and a polymer gel for serum separation or spray-coated with K2-EDTA for peripheral blood mononuclear cell (PBMC) isolation. Both 15–20 mL blood samples from each animal were immediately used for serum separation and isolation of PBMCs of the samples. The serum samples were separated from the whole blood of the dairy cows. In brief, the whole blood samples were left for 30 min at room temperature (20–25 °C) and centrifuged for 15 min at 1600 g at 4 °C, and the upper supernatant was harvested into 1.5 mL tubes. All serum samples were immediately stored at –80 °C until biochemistry analysis. Whole blood from the K2-EDTA tubes was used for PBMC isolation. For PBMCs isolation, 1:1 phosphate-buffered saline (PBS)-diluted whole-blood samples were added to Lymphoprep (STEMCELL Technologies lnc., Vancouver, BC, Canada). After centrifugation for 30 min at 800 g without braking and at 22 °C, PBMCs were harvested. For purification, PBMCs were washed twice with PBS. 

### 2.4. Biochemistry Analysis

The biochemistry analysis was conducted on a total of 18 items (protein, albumin, glucose, triglyceride, total cholesterol, low-density lipoprotein cholesterol (LDL cholesterol), blood urea nitrogen (BUN), creatinine, alanine aminotransferase (ALT), aspartate aminotransferase (AST), gamma-glutamyltransferase (GGT), creatine kinase (CK), calcium (Ca), phosphorus (P), sodium (Na), potassium (K), chlorine (Cl), and magnesium (Mg)) using the dairy cow serum samples. Under normal and high-heat environmental conditions, the Holstein and Jersey samples were used (normal environmental conditions: 8 Holsteins and 7 Jerseys; high-heat environmental conditions: 8 Holsteins and 7 Jerseys). All blood parameters were measured by each method (LABOSPECT 008AS, Hitachi, Japan) and processed by the following methods: colorimetric assay (protein, albumin, Ca, Mg), enzymatic (AST, ALT, GGT, total cholesterol, triglyceride), ion-selective electrode (P, Na, K, Cl), UV assay (CK, glucose), urease/GLDH (BUN), kinetic colorimetric (creatinine), and the homogeneous enzymatic colorimetric test (LDL cholesterol). All analyses were performed by Seegene Inc. (Seoul, Korea).

### 2.5. Flow Cytometry Analysis of Dairy Cow PBMCs

For immune cell population quantification, the 7 Holstein and 7 Jersey cows’ PBMCs were used for normal environmental conditions, and 8 Holstein and 7 Jersey cows’ PBMCs were used for high-heat environmental conditions using flow cytometry (FACS Canto II; BD Biosciences, Heidelberg, Germany). Briefly, PBMCs were stained with the following direct fluorescence-conjugated antibodies: anti-CD4:Alexa Fluor 647 (Bio-Rad, MCA1653A647), anti-CD8:Alexa Fluor 647 (Bio-Rad, MCA837A647), anti-CD11b:FITC (Bio-Rad, Hercules, California, USA; MCA1425F), anti-CD21:PE (Bio-Rad, MCA1424PE), anti-CD172a:PE-Cy5 (Bio-Rad, MCA2041C), anti-WC1:FITC (Bio-Rad, MCA838F), and anti-MHCII:FITC (Bio-Rad, MCA5656F). All conjugated antibodies were diluted 1:100 in PBS and incubated for 30 min in dark conditions. PBMCs were then fixed using 4 % paraformaldehyde (PFA) and stored at 4 °C until analysis. The flow cytometry analysis was conducted by three panel groups: (1) CD4, CD21, and WC1 antibodies for CD4 T cells, γδ T cells, and B cells; (2) CD11b and CD172a antibodies for monocytes; and (3) CD8 and MHCII antibodies for CD8 T cells and dendritic cells. 

### 2.6. Statistical Analysis

The statistical analyses were conducted using Prism 8 software (GraphPad, La Jolla, CA, USA). A one-way analysis of variance (one-way ANOVA) was used for the monthly average of temperature, relative humidity, and THI. A repeated measures two-way ANOVA was used for analysis of the respiration rate and rectal temperature of dairy cows. A mixed-effects model was used for biochemistry and flow cytometry analysis. All statistical analyses were conducted by Tukey’s multiple comparison test. In the repeated measures two-way ANOVA and mixed-effects model, the effect of breed (Holstein or Jersey), environment (normal environmental conditions or high-heat environmental conditions), or the breed × environment interaction were analyzed. All data are presented as means ± SD, and *p* < 0.05 was considered statistically significant and *p* < 0.1 was considered to have a tendency.

## 3. Results

### 3.1. Changes in Environmental Conditions Based on Temperature–Humidity Index

THI, which is derived from temperature and humidity, is a major index for studying heat stress responses. We monitored THI for several months (Figure 1). The temperature increased from April to August, and relative humidity (RH) increased from May to August. The average monthly THI was 62.9 ± 5.5 (average 17.5 °C and 59.7% RH) in April and 72.4 ± 4.2 (average: 25.0 °C and 55.7% RH) in May. The average THI in July was 80.2 ± 3.4 (average: 28.6 °C and 77.8% RH), and the average THI in August was 83.4 ± 4.0 (average: 30.3 °C and 80.5% RH). Based on the THI value that was derived from environmental condition records, we conducted the experiment using May as the normal environmental condition and August as the high-heat environmental condition [20]. The weekly average THI for the normal environmental condition was 67.8 ± 5.0 (average: 21.7 °C and 61.2% RH), and the weekly average THI for the high-heat environmental condition was 87.3 ± 1.3 (average: 32.8 °C and 79.4% RH). The Holstein and Jersey cows were exposed to normal and high-heat environmental conditions for at least 7 consecutive days. We collected blood samples under normal (May 3rd, THI = 69.6, 24.4 °C, and 36.3% RH) and high-heat (Aug 9th, THI = 87.5, 35.5 °C, and 59.6% RH) environmental conditions on daytime.

### 3.2. Measurement of Respiration Rate and Rectal Temperature of Dairy Cows 

The respiration rate and rectal temperature are considered reliable parameters for determining the degree of heat stress in dairy cows. The respiration rates and rectal temperatures of the two breeds of dairy cows under normal and high-heat environmental conditions were monitored. Under normal environmental conditions, the Holstein and Jersey cows did not exhibit significantly different levels of physiological heat stress responses. In high-heat environmental conditions, the two breeds of cows had significantly increased respiration rates (Figure 2A), but there was no significant difference in respiration rate between the Holstein and Jersey cows in these conditions. However, in the heat stress environment, only Holstein cows showed increased rectal temperature. This means that Jersey cows showed a tendency of reduced rectal temperature compared with Holstein cows (*p* < 0.1) (Figure 2B). These results indicate that, under high-heat environmental conditions, Jersey cows showed a milder heat stress response than Holstein cows. After verification of this distinct phenotypic heat stress response, our study focused on understanding the immuno-physiological changes in the two dairy cow breeds.

### 3.3. Changes in the Biochemistry Analysis of Two Breeds of Dairy Cows under Different Enviromental Conditions

The biochemistry analysis for three categories (biochemical metabolites, serum enzymes, and mineral contents), including a total of 18 items, was conducted under normal and high-heat environmental conditions. For the biochemical metabolites, there was no difference between breeds in both the normal and high-heat experimental conditions (Figure 3). The protein, albumin, glucose, total cholesterol, LDL cholesterol, and BUN were significantly decreased by heat stress in both the Holstein and Jersey cows (*p* < 0.05). Triglyceride only showed a decrease in the Holstein cows. The four serum enzymes, ALT, AST, GGT, and CK, were assessed in the two breeds of dairy cows under different environmental conditions (Figure 4). The concentrations of ALT in dairy cow blood samples tended to be higher in the Holstein cows than in the Jersey cows under the normal environmental conditions (*p* = 0.0503). This was significantly decreased in the Holstein cows under heat stress (*p* < 0.05); however, the Jersey cows showed no change in ALT between the normal and high-heat environmental conditions. There was no difference in AST for both breeds of cow between normal and high THI conditions. Moreover, the concentrations of AST were decreased only in the Jersey cows under heat stress conditions (*p* < 0.05). Heat stress resulted in decreased concentrations of CK in both the Holstein and Jersey cows significantly or in a tendency. There was no change in GGT concentrations depending on the environmental conditions and breed. The mineral contents in the dairy cow sera showed changes due to heat stress but no difference by breed (Figure 5). The concentrations of Ca, Na, K, Cl, and Mg decreased or showed a decreasing tendency under heat treatment in both the Holstein and Jersey cows. The concentrations of P were decreased by heat stress in the Holstein cows but not in the Jersey cows.

### 3.4. Alteration in the Population of Peripheral Blood Mononuclear Cells in Dairy Cows by Heat Stress

The immune cell population of PBMCs in dairy cows was assessed by flow cytometry analysis under normal and high-temperature environmental conditions. B cells and monocytes were significantly or had a tendency to be altered by heat stress (Figure 6 and Figure 7). 

The B cell (CD4–CD21+) populations were increased in Holstein PBMCs under heat stress (*p* < 0.05). The population of monocytes (CD11b+CD172a+) of PBMCs showed a decreased tendency under heat stress in the Holstein cows (*p* < 0.1) (Figure 7).

Some types of T cells in PBMCs, CD4 T cells (CD4+WC1–) and γδ T cells (CD4–WC1+), were not altered by heat stress; however, there was a different trend in T cell subset populations between the Holstein and Jersey cows (Figure 8A). The CD4 T cell ratio in Holstein cows had a higher tendency than that in the Jersey cows under normal temperature conditions (*p* < 0.1). The CD8 T cells (CD8+MHCII–) and dendritic cells (CD8+MHCII+) in dairy cow PBMCs were not significantly different between environments or breeds under normal and hyperthermic conditions (Figure 8B).

## 4. Discussion

Heat stress directly induces multiple physiological changes, including metabolism [22] and immunity [23,24]. Heat-stressed dairy cows usually show the common symptoms of heightened respiration rate, panting, sweating, and water intake, and increased rectal temperature [25]. The THI, used as an indicator of heat stress conditions, was first developed by Thom [26] and sequentially modified by Kibler [27], Yousef [28], and Mader, Davis, and Brown-Brandl [29]. The THI is divided into five categories based on air temperature and humidity. The ranges of THI represent the degree of heat stress in dairy cows: THI < 71: comfort; THI < 72 to <79: mild heat stress; THI < 80 to <89: moderate heat stress; THI > 90 severe heat stress; THI > 100: death [20]. For Holstein cows under normal conditions, the rectal temperature range was 38.4–38.8 °C when the THI ranged from 60 to 72 [30,31]. By contrast, in the heat stress conditions, with a THI range of 72–89, the rectal temperature range was 39.0–39.1 °C [32,33]. However, in a heat stress environment, the Jersey cows displayed a milder heat stress response, with rectal temperatures ranging from 37.2 to 38.7 °C in a THI range of 70–80 [34]. In our study, the average THI in the first phase of the experiment was 67.8 ± 2.1 (average: 21.7 °C and 61.2% RH), which fits into the comfort range. The average THI in the second phase of the experiment was 87.3 ± 0.5 (average: 32.8 °C and 79.4% RH), which fits into the moderate heat stress range. In addition, both Holstein and Jersey cows in the heat stress conditions showed heat stress symptoms such as increased respiration rates. Notably, Jersey cows showed a milder heat stress response than Holstein cows, because rectal temperature increased significantly in Holstein cows but not in Jersey cows; Holstein cows generally had an approximately 0.3 °C higher rectal temperature than Jersey cows under heat stress conditions. It has been previously reported that Jersey cows are more resistant to heat stress than Holstein cows [35]. Under heat stress conditions, the decline in milk yield for Holstein cows was more rapid than that for Jersey cows [15,35,36]. Collectively, the physical heat stress responses of Holstein and Jersey cows were not significantly different under normal environmental conditions, although they showed significant differences in the heat stress environmental conditions.

Heat stress can induce a reduction in nutrient intake, indirectly or directly affecting the metabolism of dairy cows. For example, reduced nutrient intake indirectly accounts for approximately 35 % decreased milk yield in Holstein cows [26]. Heat-stressed dairy cows do not display the typical metabolic profile with lower nutrient status, which is a direct effect of nutrient reduction by heat stress [6]. To understand the mechanism of heat stress-induced metabolic alteration, many studies of dairy cows have been conducted using blood parameter analysis, such as metabolites, enzymes, and minerals.

In our study, glucose, total cholesterol, LDL cholesterol, protein, albumin, BUN, CK, Ca, Na, K, Cl, and Mg were decreased by heat stress in both Holstein and Jersey cows, significantly or in a tendency. These results indicate that these reductions may be a common metabolic alteration for dairy cows, regardless of breed, in a high-temperature environment. Glucose, total cholesterol, and LDL cholesterol are major metabolites of energy metabolism. The reduction in blood glucose during hot weather conditions can be explained by a reduction in energy intake. Additionally, this can also be induced by the increased cost of thermoregulation and the downregulation of gluconeogenesis, as an endocrine acclimation to heat stress conditions [37]. Two types of cholesterol reduction were observed in the hot and humid environment in Holstein and Jersey cows, which could be due to an increase in lipid utilization by other peripheral tissues. Another key metabolism in dairy cows is that of protein and related metabolites and BUN, which are reduced by heat stress in dairy cows. It is possible that skeletal muscle protein is mobilized to the blood during the lactation period [38]. As a result, reduced blood protein and BUN are also indicative of a decrease in the mobilization of protein from muscle in a high-temperature environment. Minerals play a key role in maintaining normal physiological functions in animals, including dairy cows [39]. Heat stress can increase mineral loss as well as body fluid excretion in dairy cows. Na and Ca are critical minerals in dairy cows. Na is a primary extracellular cation in animals, and its main functions include the maintenance of body fluid balance, osmotic pressure, and acid balance. It is also essential for the active transport of glucose, a major energy metabolism-related metabolite, across the plasma membranes [40]. In dairy cows, Na deficiency has been associated with loss of appetite and decreased milk yield [41]. Thus, we postulate that negative energy balance and milk production may be associated with a decrease in Na by heat stress. During hot and humid weather, Ca was decreased in both breeds of dairy cow used in this study. There are a number of putative mechanisms by which calcium deficiency status in the blood causes bovine health problems, including a reduction in uterine contractility, increased risk of negative energy balance, and suppressed immune function [42]. Thus, the reduction in these minerals caused by heat stress makes dairy cows vulnerable to various diseases, including metabolic and infectious disorders.

In the various biochemistry analysis results, alterations in some items were breed-specific. For example, the triglyceride, creatinine, ALT, and P concentrations were reduced by heat stress only in Holstein dairy cows but not in Jersey dairy cows. Under hot and humid weather conditions, high-producing dairy cows usually experience a negative energy balance as a consequence of insufficient feed intake to meet the energy requirements of milk production. The deficit between energy consumed and energy expended is primarily covered by the mobilization of fatty acids from adipose tissue, which causes increased concentrations of non-esterified fatty acids (NEFA) in the blood [43]. Some of the NEFA are converted to triglycerides to export to very low-density lipoprotein (VLDL) in the liver [44]. However, the ability of ruminants, including dairy cows, is limited to the secretion of triglycerides as part of the VLDL, leading to an accumulation of triglycerides in the liver [45]. In heat-stressed Holstein cows, the reduction in triglycerides may lead to accumulation of triglycerides from the blood in the liver, and this phenomenon can induce metabolic diseases. Like other protein metabolism metabolites, creatinine was decreased in heat-stressed Holstein cows. Creatinine in the blood is considered an index of muscle mass and, during heat stress, an alteration in creatinine could be related to the mobilization of protein from muscle [38]. Collectively, among the blood serum enzymes, ALT was significantly decreased in only Holstein cows under heat stress, and there was a difference between Holstein and Jersey cows under normal weather conditions. The blood concentrations of enzymes including AST, GGT, and ALT are regarded as hepatic function indicators [46] and are also correlated with metabolic diseases such as ketosis [47]. The blood concentrations of ALT between Holstein and Jersey dairy cows may be linked to metabolic differences and disease vulnerability in both breeds of dairy cow under different environmental conditions. Under the high-temperature and humid conditions, AST and K were significantly reduced only in Jersey cows and not Holstein cows in this study. Blood AST is also a known liver function indicator [46], and Srikandakumar et al. reported that AST was decreased by heat stress, though it was within physiological limits [48]. The lower blood K concentrations in dairy cows in high-temperature and humid environments may be related to the loss of this electrolyte through sweat [49]. Our study suggests that various blood parameters of both breeds of dairy cow were markedly changed, and some of them had breed-level characteristics.

We analyzed the immune cell population in PBMCs using flow cytometry. The immune cell populations of dairy cows have been reported to undergo changes in heat stress conditions [50]. Under hot weather conditions, the B cell population of Holstein cows was significantly increased in August (high-heat environmental conditions) compared with that in May (normal environmental conditions), but Jersey cows did not show significant differences between May and August. The lymphocytes are composed of B cells and T cells and are involved in various immunological functions, such as immunoglobin production and modulation of the immune response by cytokine production [51]. Naik et al. [52] reported that lymphocytes were significantly higher during summer than during other seasons in Punganur cattle. However, many studies have shown that the lymphocyte population decrease during the summer season or under heat stress [53,54,55]. Differences among these studies might be caused by the difference in breed and/or heat stress conditions. This study provides evidence that humoral immune system function, such as immunoglobulin synthesis of dairy cows, can be affected by a changing B cell population under heat stress conditions.

We compared CD4, CD8, and γδ T cells between Holstein and Jersey dairy cows in normal and hot and humid environments. As a result, the CD4 T cell populations differed between the two breeds of dairy cow in a normal environment (*p* < 0.1); however, the population of CD8 T cells did not differ depending on the breed and environmental conditions. The proportion of CD4+ T cells (%) in Holstein cows presented a higher tendency than that in Jersey cows in May. Murine and human CD4+ T cells are further subdivided into helper T cell subsets such as Th1, Th2, Th17, and Treg cells according to their cytokine expression patterns. Th1 cells are involved in the elimination of intracellular pathogens, and Th2 cells mount immune responses to extracellular helminth and play a major role in allergic responses [56]. Th17 cells play critical roles in the development of autoimmunity and allergic reactions [57]. The properties of Treg cells are anti-inflammatory and maintain tolerance against self-components [58]. In this regard, the difference in the proportion of normal-condition CD4+ T cells between Holstein and Jersey dairy cows may have differential immune characteristics.

Heat-stressed Holstein cows showed a decreasing tendency of monocyte populations in August compared with those in May. Monocytes are myeloid cells produced by the bone marrow that play important roles in the immune response, such as to infections and injuries [59]. Contrary to the results of this study, another study reported that monocyte counts increased in summer [60]. In this regard, this study demonstrated decreases in monocytes under heat stress conditions or during the summer period in Holstein cows.

In the dairy cow industry, heat stress is a major factor reducing productivity. Heat stress can cause a decrease in DMI, which leads to a nutritional imbalance in dairy cows. Heat-stressed dairy cows generally have weakened immunity [50]. In this study, we observed various alterations in blood metabolites and the PBMC population in the two breeds of dairy cow during high temperatures in a humid weather environment. The blood metabolites also related to white blood cells (WBCs). For example, WBCs directly varied with blood betahydroxylbutyrate (BHBA) or glucose in Holstein cows [61]. These relationships reflect the connection between nutrition status and immunity. Immuno-physiological responses can be altered by many factors, such as genetic background, use of nutrients, environmental conditions, and aging [62]. Recently, we reported heat stress-induced ruminal microbiome changes in two breeds of dairy cow [21]. Various studies have suggested that metabolites have an important impact on the immune response in animals, including dairy cows [5]. Thus, further studies need to be conducted to understand the mechanism by which altered metabolites regulate blood immune cell functions in dairy cows. Undoubtedly, lactation number and period can significantly affect dairy cow nutrition and immunity. We performed an experiment using Holstein and Jersey cows with similar parity and DIM. In this study, we used that same dairy cow herd that had similar DIM (Holstein: 139 ± 8.0; Jersey: 128 ± 15.0), mid-lactation period, parity number (Holstein: 2.4 ± 0.5; Jersey: 2.4 ± 0.5), and ages (Holstein: 52 ± 7.9 months old; Jersey: 57 ± 9.0 months old). However, we cannot fully exclude of various factors such as aging and lactation period, as this study was conducted over a 3-month interval. Another point which needs to gain attention is that Holstein and Jersey cows have differential nutritional requirements for the optimal level of milk production. In this study, we offered feed on a limited amount basis according to the farm feeding program. Holstein and Jersey cows were fed the same amount of TMR that exceeded the level of the nutritional requirements for DE and ME based on the Korean Feeding Standard for dairy cattle (2017) [18] for mid-lactation cows (milk yield: 30 kg, milk fat: 4.3%, milk protein: 3.4%, ME: 52.0 Mcal/day). All experimental cows were fed TMR with no residue. This also indicates that the nutritional level of the feed may not completely meet the nutrient requirements of all dairy cows used in the study, especially Holstein cows, under normal environmental conditions. However, this usually occurs in raising dairy cows on farms using a limited feeding strategy, as every cow has different milk production and nutritional requirements. Therefore, our study presents natural changes in immuno-physiological responses in each Holstein and Jersey dairy cow in the mid-lactation period between normal and heat stress environments.

To better understand the direct impact of heat stress on dairy cows, specific experimental methods should be used, such as a climate control chamber system or an electric blanket, for dairy cow studies [63,64]. To elucidate the functional outcome from heat stress-induced impaired immune response, immune challenges model could be applied for future study.

## 5. Conclusions

In this study, the effects of heat stress on the metabolism and PBMC population in Holstein and Jersey dairy cows were examined by biochemistry analysis and flow cytometry analysis. In the biochemistry analysis of the dairy cow sera, both breeds showed decreased concentrations of metabolites (protein, albumin, glucose, total cholesterol, LDL cholesterol, BUN, CK, Ca, Na, K, Cl, and Mg). Three blood parameters were decreased in Holstein cows only (triglyceride, ALT, and P). In heat-stressed Jersey cows, the concentrations of AST decreased. Through biochemistry analysis, we confirmed that heat-stressed dairy cows have reduced energy-related metabolites. We also identified the differential immune cell population of PBMCs by breed and environment. In heat-stressed Holstein cows, the proportion of peripheral blood B cells were significantly increased, however, the monocytes showed a decreasing tendency. Among the PBMCs, there was also a difference in T cells. The T cell subset did not change in the heat stress conditions when compared with that in the normal conditions in both breeds of dairy cow. Under normal environmental condition, CD4+ T cells presented a lower tendency in Jersey cows than in Holstein cows. The major findings of this study provide key information regarding metabolism and immune cell population changes in two breeds of dairy cow under heat stress.

## Figures and Tables

**Figure 1 animals-11-00974-f001:**
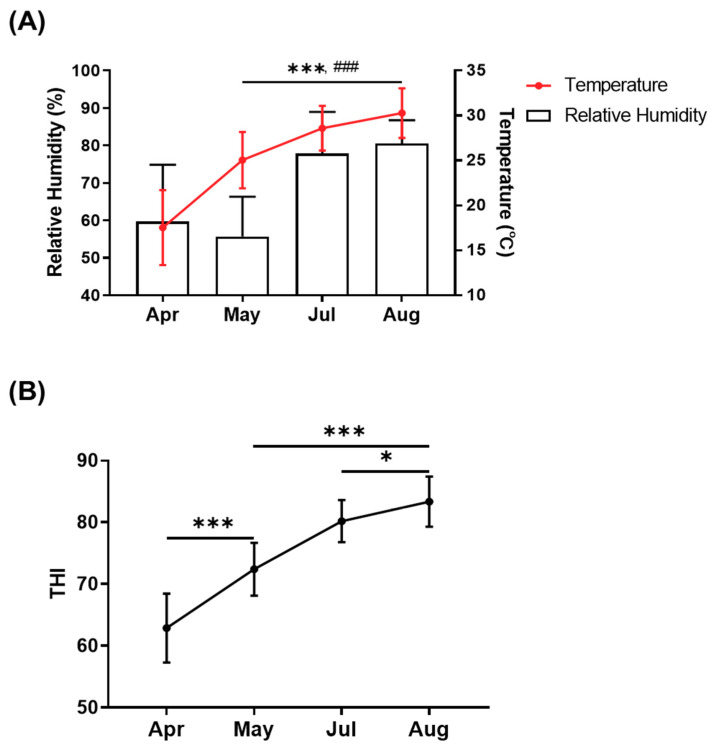
Changes in the temperature–humidity index (THI) in this study. Monthly average of temperature and relative humidity (**A**). Monthly average of THI derived from temperature and humidity (**B**). Data are represented as means ± standard deviation (SD). Values were statistically analyzed by one-way ANOVA with Tukey’s multiple comparison test. * *p* < 0.05, *** *p* < 0.001, and ### *p* < 0.001. In Panel A, the asterisk (* and ***) means a significant difference in temperature, and the number sign (###) means a significant difference in relative humidity.

**Figure 2 animals-11-00974-f002:**
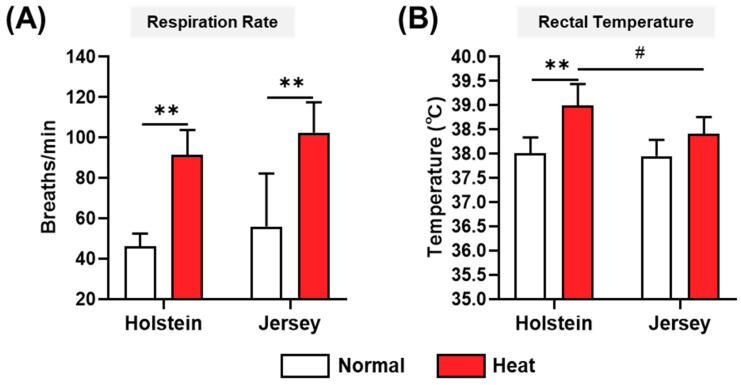
Physical heat stress parameters of two breeds of dairy cow under different environmental conditions. Measurements of respiration rate (**A**) and rectal temperature (**B**) for Holstein and Jersey cows under normal environmental conditions (Normal, THI = 69.6) and high-heat environmental conditions (Heat, THI = 87.5). Data are represented as means ± standard deviation (SD); *n* = 8 animals/group. Values were statistically analyzed by two-way repeated measure ANOVA with Tukey’s multiple comparison test. ** *p* < 0.01 and # *p <* 0.1.

**Figure 3 animals-11-00974-f003:**
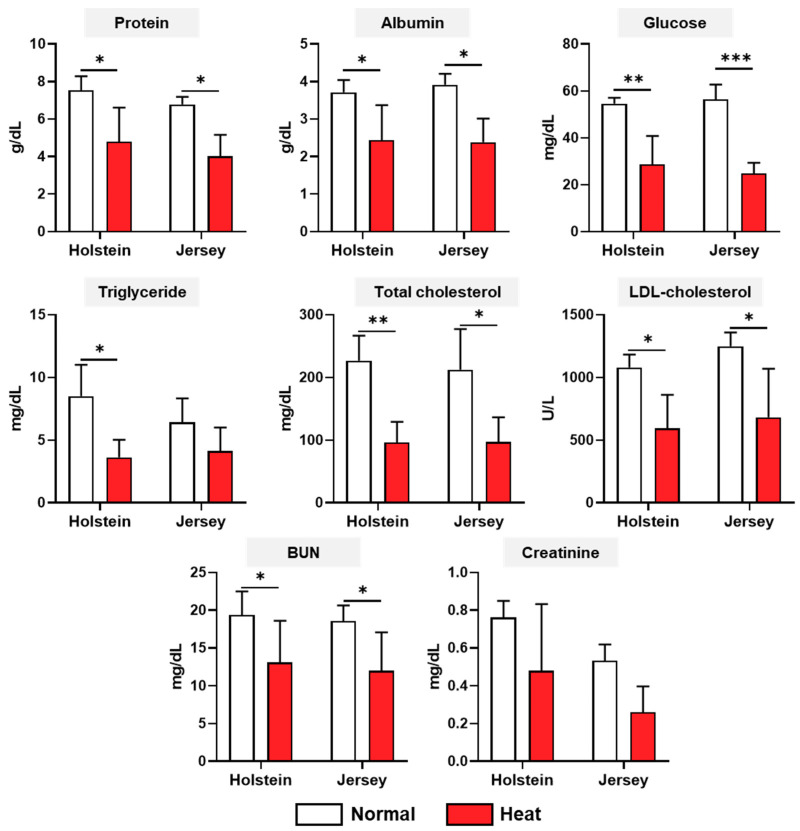
Changes in biochemical metabolites in two breeds of dairy cow under heat stress. Biochemistry analysis of Holstein and Jersey cows was conducted for blood biochemical metabolites under normal and high-heat conditions. Data are represented as means ± standard deviation (SD); *n* = 7–8 animals/group. Values were statistically analyzed by a mixed-effects model with Tukey’s multiple comparison test. * *p* < 0.05, ** *p* < 0.01, and *** *p* < 0.001.

**Figure 4 animals-11-00974-f004:**
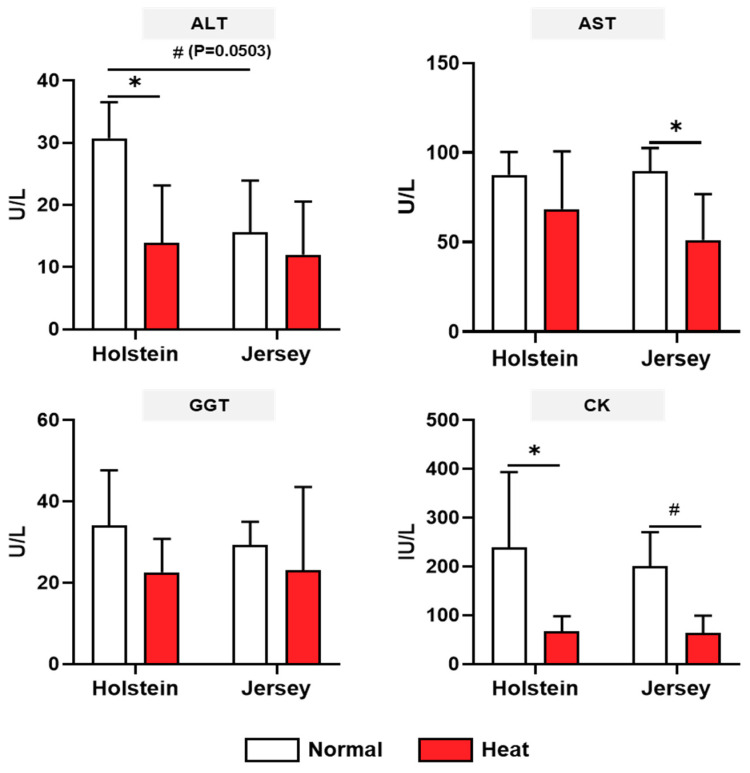
Changes in serum enzyme concentrations from two breeds of dairy cow under heat stress. Biochemistry analysis of Holstein and Jersey cows was conducted for serum enzymes under normal and high-heat conditions. Data are represented as means ± standard deviation (SD); *n* = 7–8 animals/group. Values were statistically analyzed by a mixed-effects model with Tukey’s multiple comparison test. * *p* < 0.05 and # *p <* 0.1; ALT, alanine aminotransferase; AST, aspartate aminotransferase; GGT, gamma-glutamyltransferase; CK, creatine kinase.

**Figure 5 animals-11-00974-f005:**
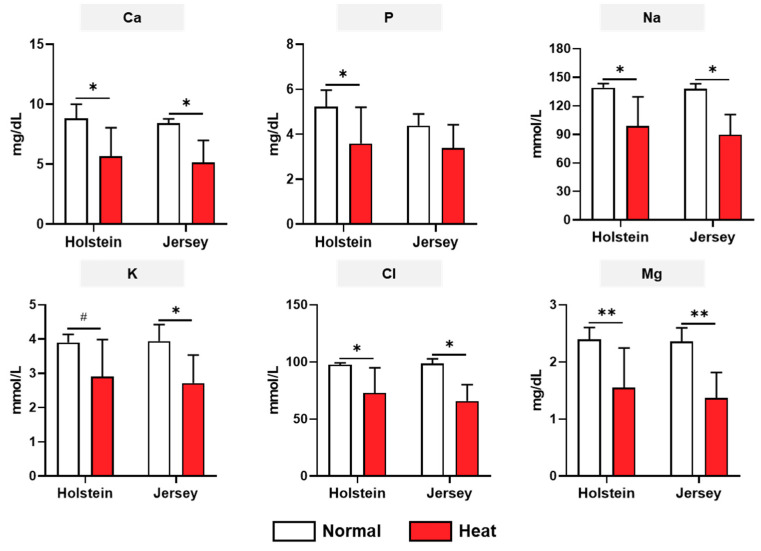
Changes in mineral concentrations in dairy cows under heat stress. Biochemistry analysis of Holstein and Jersey cows was conducted for mineral contents under normal and high-heat conditions. Data are represented as means ± standard deviation (SD); *n* = 7–8 animals/group. Values were statistically analyzed by a mixed-effects model with Tukey’s multiple comparison test. * *p* < 0.05, ** *p* < 0.01, and # *p <* 0.1.

**Figure 6 animals-11-00974-f006:**
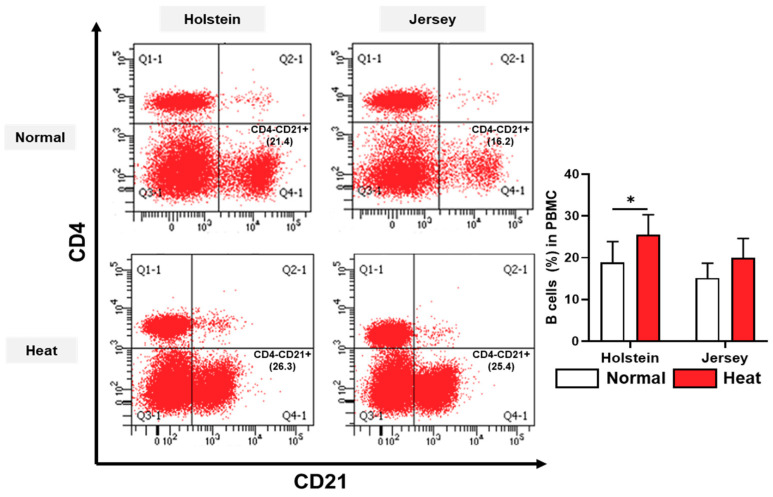
Alteration in blood B cells in peripheral blood mononuclear cells (PBMCs) from two breeds of dairy cow under different environmental conditions. The populations of B cells (CD4-CD21+) in Holstein and Jersey cows under normal and high-heat environmental condition were analyzed using flow cytometry. Data are represented as means ± standard deviation (SD); *n* = 7–8 animals/group. Values were statistically analyzed by a mixed-effects model with Tukey’s multiple comparison test. * *p* < 0.05.

**Figure 7 animals-11-00974-f007:**
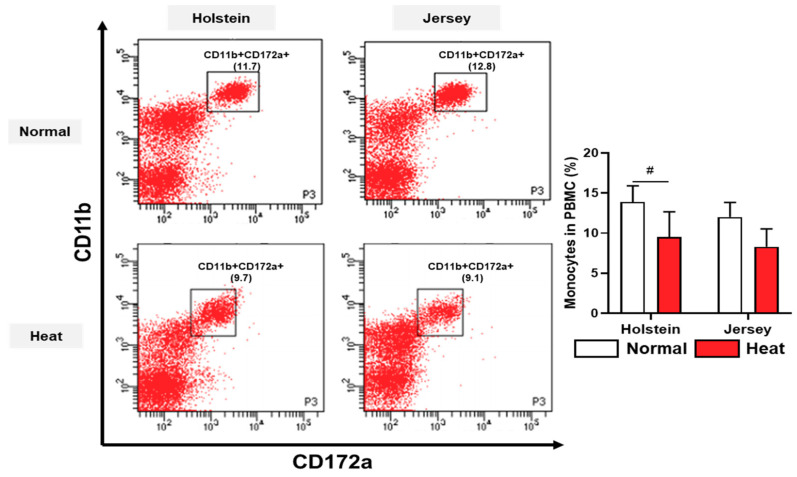
Alteration in blood monocytes in PBMCs from two breeds of dairy cow under different environmental conditions. The populations of monocytes (CD11b+CD172a+) in Holstein and Jersey cows under normal and high-heat environmental conditions were analyzed using flow cytometry. Data are represented as means ± standard deviation (SD); *n* = 7–8 animals/group. Values were statistically analyzed by a mixed-effects model with Tukey’s multiple comparison test. # *p* < 0.1.

**Figure 8 animals-11-00974-f008:**
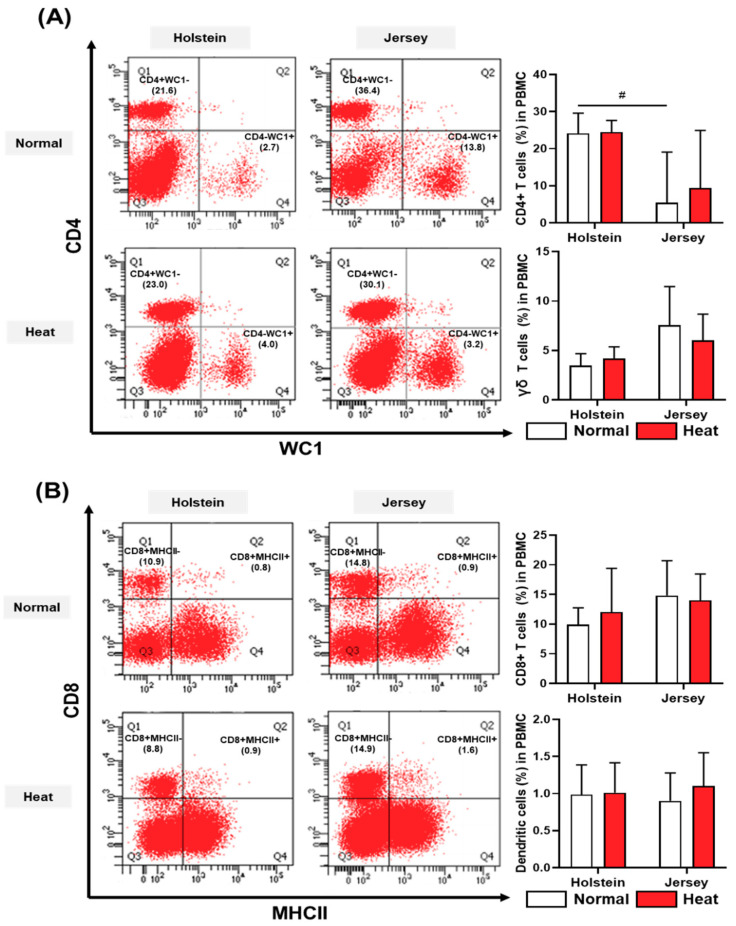
Alteration in blood T cells and dendritic cells in PBMCs from two breeds of dairy cow under different environmental conditions. The population of T cells and dendritic cells in Holstein and Jersey cows under normal and high-heat environmental conditions were analyzed using flow cytometry. The CD4 T cell (CD4+WC1–) and γδ T cell (CD4–WC1+) populations in PBMCs are presented in (**A**), and CD8 T cell (CD8+MHCII-) and dendritic cell (CD8+MHCII+) populations in PBMCs are shown in (**B**). Data are represented as means ± standard deviation (SD); *n* = 7–8 animals/group. Values were statistically analyzed by a mixed-effects model with Tukey’s multiple comparison test. # *p* < 0.1.

**Table 1 animals-11-00974-t001:** Ingredients and nutrients of the experimental diets used in this study.

Item	Amount
Ingredients composition, % of DM	-
Concentrate	15.3
Soybean meal	2.4
Corn silage	47.2
Alfalfa hay	7.1
Tall fescue	9.4
Timothy	5.9
Energy booster ^†^	7.1
Cash Gold ^†^	4.5
Lyzin-Plus ^‡^	0.2
Limestone	0.2
Zin Care ^†^	0.1
Supex-F ^†^	0.5
Trace minerals ^§^	0.05
Vitamin premix^¶^	0.05
Chemical composition	-
Dry matter (DM), %	53.2
Crude protein, % of DM	16.6
Neutral detergent fiber, % of DM	37.0
Acid detergent fiber, % of DM	25.6
Calcium, % of DM	0.4
Phosphorus, % of DM	0.15
Total digestible nutrient, %	69.6
Digestible energy, Mcal/kg	3.069
Metabolizable energy, Mcal/kg	2.650

^†^ Cofavet, Cheonan, Republic of Korea. ^‡^ A.N.Tech, Cheonan, Republic of Korea. ^§^ Contained 0.40% Mg, 0.20% K, 4.00% S, 0.08% Na, 0.03% Cl, 400 mg of Fe/kg, 60,042 mg of Zn/kg, 16,125 mg of Cu/kg, and 42,375 mg of Mn/kg. Provided approximately 5000 KIU of vitamin A/kg, 1000 KIU of vitamin D/kg, 33,500 mg of vitamin E/kg, and 2400 mg of vitamin C/kg.

## Data Availability

All data presented in this study are available on request from the corresponding authors.

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
