# Peer review of "Changes in Blood Metabolites and Immune Cells in Holstein and Jersey Dairy Cows by Heat Stress"

_animals, 2021, doi:10.3390/ani11040974_

Round 1

Reviewer 1 Report

The manuscript by Joo et al describes the effects of breed on immunity and metabolism in response to heat stress. This is an area of great interest for the dairy industry. However, the manuscript lacks a significant amount of detail within the Materials & Methods section which make interpretation of the results difficult. Please see general and specific comments below.

General Comments:

  • The Materials & Methods section lacks detail. The readers are unable to understand how the cows were exposed to heat stress, how many cows experienced heat stress and for how long, and when blood samples were collected relative to heat stress exposure. Due to this, there is no way to make appropriate inferences based on the data as it is unclear how the cows were exposed and when the samples were collected.
  • Is there any information on weight change, feed or water intake, or milk yield in the cows during this study? This information would strengthen the manuscript and would provide some context for the changes observed in the metabolites measured.

Specific comments:

Should metabolism be included as a key word?

Lines 72-74: How were the leukocyte populations altered?

Line 78-79: Please include a reference for this statement.

Line 81: …having a 0.3 oC greater body temperature…

Line 103: Please include average days in milk for both breeds.

Line 103-105: Where were the tie stalls located? Was this an enclosed barn? Please provide more details of the housing of the cows.

Line 109: How was water provided to the cows? Ad libitum? Shared water? Please provide more details.

Lines 116-118: For clarification, this was a naturally occurring heat stress incident, not an induced heat stress condition? Please add more information. What dates did the study include? Was this only for one week? With this limited amount of information it is hard to understand how the heat stress was imposed on the cows. How many cows were experiencing heat stress and how many experiencing thermal-neutral temperatures?

Line 126: When were these samples collected and how frequently? Were cattle restrained for the collection of the samples?

Line 132: Were serum samples allowed to clot for a period of time before centrifugation?

Line 138: Was a red blood cell lysis step included?

Line 140: Please list the 18 variables measured.

Line- 145-147: This is the first mention of how many cows used under different conditions. However, were the same cows used under different temperature conditions as the beginning of the Materials & Methods section lists only 8 of each breed were used. What is included here would suggest 29 cows were used. Please clarify.

Line 147: Were PBMC samples analyzed immediately after isolation? If not, please describe storage conditions used prior to analysis on the flow cytometer.

Line 172: Based on this information, was this a retrospective analysis based on environmental readings? Please clarify and provide more of this information in the Materials & Methods section.

Line 176: This information needs to be in the Materials & Methods section. How was it determined when to collect these samples? When were the samples collected? More information needs to be included.

Line 195: Please rephrase this sentence for clarity. Use ‘reduced’ rather than ‘lower’.

Line 209-214: Please include this information in the Materials & Methods section rather than in the Results section.

Line 221: Please do not start a sentence with an abbreviation. Additionally, use ‘concentrations’ rather than ‘levels’ and ‘greater’ rather than ‘higher’ here and throughout.

Are there any indications in the literature to support whether the changes observed in some of the metabolites may be related to the changes observed in leukocyte populations?

Line 356-357: Please rephrase this sentence for clarity.

Line 392: An increase in B cells does not necessarily mean there is an increase in the humoral immune system. A better test of this would be the measurement of immunoglobulins.

Author Response

Response to reviewer 1

Comments and Suggestions for Authors: The manuscript by Joo et al describes the effects of breed on immunity and metabolism in response to heat stress. This is an area of great interest for the dairy industry. However, the manuscript lacks a significant amount of detail within the Materials & Methods section which make interpretation of the results difficult. Please see general and specific comments below.

General Comments

The Materials & Methods section lacks detail. The readers are unable to understand how the cows were exposed to heat stress, how many cows experienced heat stress and for how long, and when blood samples were collected relative to heat stress exposure. Due to this, there is no way to make appropriate inferences based on the data as it is unclear how the cows were exposed and when the samples were collected.

Response: Thanks for valuable comments. We agree that there are lack detailed information for experimental procedure and design in M & M section. Overall, we added essential information to make readers to understand easily in revised manuscript. Please find information: how the cows were exposed to heat stress and blood sample collection (Line 132-137; 141-147).

Is there any information on weight change, feed or water intake, or milk yield in the cows during this study? This information would strengthen the manuscript and would provide some context for the changes observed in the metabolites measured.

Response: Thanks for valuable comments. We agree that physiological changes including feed intake, weight, milk yield changes closely associated with metabolic profiles in heat stressed cows. We added some more information such as parity number, days in milk, somatic cell count, housing of the dairy cows in revised manuscript. Unfortunately, we could not assess monitoring of body condition score change measurement in our experimental setting. (Line 97-118)

Specific comments:

Should metabolism be included as a key word?

Response: Thank you for kind suggestion. We have added “Metabolism” as a key word.

Lines 72-74: How were the leukocyte populations altered?

Response: We have changed to “decreased from “altered” to make clear. (Line 74)

Line 78-79: Please include a reference for this statement.

Response: We have added a reference for this sentence. (Line 81)

(West, J.W. Effects of heat-stress on production in dairy cattle. J. Dairy Sci. 2003, 86, 2131–2144)

Line 81: …having a 0.3 oC greater body temperature…

Response: Thank you for correction. We have revised “higher” to “greater”. (Line 82)

Line 103: Please include average days in milk for both breeds.

Response: We added DIM information for Holstein and Jersey cow used in this study. (Line 107-109)

Line 103-105: Where were the tie stalls located? Was this an enclosed barn? Please provide more details of the housing of the cows.

Line 109: How was water provided to the cows? Ad libitum? Shared water? Please provide more details.

Response: We added information for cow hosing and feeding in revised manuscript. (Line 110-118)

Lines 116-118: For clarification, this was a naturally occurring heat stress incident, not an induced heat stress condition? Please add more information. What dates did the study include? Was this only for one week? With this limited amount of information it is hard to understand how the heat stress was imposed on the cows. How many cows were experiencing heat stress and how many experiencing thermal-neutral temperatures?

Response: Thank you for valuable comment. In this study, we focused on naturally occurring heat stress incident that changed from Normal condition to Heat condition by seasonal change. To clarify, we added “by seasonal change” in revised manuscript. (Line 125)

Our study was conducted normal environmental condition (May) and high-heat environmental condition (August) based on THI. Firstly, we monitored the monthly THI (April and May, July and August; Figure 1) and selected experimental time point May and August. Secondly, we monitored the weekly average THI for May and August to confirm the sampling date based on reference. (Normal week-April 26th to May 3rd / Heat week-August 2nd to August 9th). Finally, we collected blood of dairy cows on May 3rd as normal-environmental condition and Aug 9th as high-heat environmental condition. Cows exposed different environmental condition at least 7 constitutive days. We described this information in result M&M and (Line 132-137) section (3.1. Changes in environmental condition based on temperature-humidity index). (Line 204-215)

Line 126: When were these samples collected and how frequently? Were cattle restrained for the collection of the samples?

Response: Thank you for valuable comment. The blood samples were collected only once immediately after THI and rectal temperature measurement. Cows exposed different environmental condition at least 7 constitutive days. The dairy cows were also fixed for blood sampling using ropes. We corrected sentence for blood sampling information in revised manuscript. (Line 141-147)

Line 132: Were serum samples allowed to clot for a period of time before centrifugation?

Response: We used SST tubes for serum isolation. Before centrifugation, the blood samples were left for 30 min on room temperature. We add this information in revised manuscript. (Line 149-151)

Line 138: Was a red blood cell lysis step included?

Response: The RBC lysis step did not include in our study. We used Lymphoprep™ (Stemcell Technologies, #07801) that is density gradient medium for PBMC isolation. After centrifugation, the diluted blood samples were separated to 4 floors (Plasma layer, Buffy layer (PBMCs), Lymphoprep layer, RBCs) and we harvest only “Buffy layer” carefully. We added representative Lymphoprep™ images before and after centrifugation.

(Cockshell and Bonder, 2016)

Line 140: Please list the 18 variables measured.

Response: Sorry for our mistake. We have added whole list for variables in revised manuscript. (Line 158-162)

Line- 145-147: This is the first mention of how many cows used under different conditions. However, were the same cows used under different temperature conditions as the beginning of the Materials & Methods section lists only 8 of each breed were used. What is included here would suggest 29 cows were used. Please clarify.

Response: Sorry for confusing. Unfortunately, we lost some PBMC and serum samples during experimental progress due to technical issue. In normal environment samples (May samples), we lost two samples (1 Holstein and 1 Jersey) and lost one sample (1 Jersey) on high-heat environmental condition (Aug samples). In the biochemistry analysis of Jersey cows, we used seven serum samples both normal and high-heat environmental condition. We added exact numbers of sample for each analysis in revised manuscript, clearly. (Line 163-165; 173-175)

Line 147: Were PBMC samples analyzed immediately after isolation? If not, please describe storage conditions used prior to analysis on the flow cytometer.

Response: We added storage condition and further process in revised manuscript. (Line 176-183)

Line 172: Based on this information, was this a retrospective analysis based on environmental readings? Please clarify and provide more of this information in the Materials & Methods section.

Response: Yes, this is retrospective study. This study provides useful information for natural changes of dairy cows exposing environmental change. We added more information about experimental design and environmental condition setting in M & M section (Line 132-137; 204-215)

Line 176: This information needs to be in the Materials & Methods section. How was it determined when to collect these samples? When were the samples collected? More information needs to be included.

Response: We removed this sentence in results part and move to Materials & Methods part with more information and some correction. (Line 140-146)

Line 195: Please rephrase this sentence for clarity. Use ‘reduced’ rather than ‘lower’.

Response: Thank you for correction. We changed “lower” to “reduced” in revised manuscript. (Line 233)

Line 209-214: Please include this information in the Materials & Methods section rather than in the Results section.

Response: Thank you for comment. We added item list of biochemistry analysis in M & M and simplified these contents in results section. (Line 158-162; 246-248)

Line 221: Please do not start a sentence with an abbreviation. Additionally, use ‘concentrations’ rather than ‘levels’ and ‘greater’ rather than ‘higher’ here and throughout.

Response: Thank you for suggestion. We revised these issues carefully. (Line 254-256)

Are there any indications in the literature to support whether the changes observed in some of the metabolites may be related to the changes observed in leukocyte populations?

Response: Thank you for suggestion. We added some relevant references in revised manuscript. (Line 451-454)

Line 356-357: Please rephrase this sentence for clarity.

Response: Sorry for confusing. We corrected the sentence to “Some of the NEFA are converted to triglycerides to export to very-low density lipoprotein (VLDL) in the liver.” (Line 388-390)

Line 392: An increase in B cells does not necessarily mean there is an increase in the humoral immune system. A better test of this would be the measurement of immunoglobulins.

Response: We agree your opinion. In revised manuscript, we do not overestimate the result of B cell changes. We just mentioned that increased B cell numbers may be associated with altered humoral immunity. However, examination of immunoglobulins levels is needed to assess functions of humoral immunity in future study (Line 423-426).

Reviewer 2 Report

I think its a very complete study, with an extensive literature review and with an easy to understand lenguage for any reader, however, his contribution to new scientific knowloge is low. 

I believe that there are failures in the statistical methodology applied, because the observations of the reponse variables, are measured in the same animal during two conditions (normal and heat stress) which generates covariance between observations, which would have  been resaned using a mixed effects model, which considers the types of variance covariance matrix thas is generated, which is not achieved with SAS PROC GLM, also allows the analysis of the breed x stress condition interaction, which is important to results interpretation. 

Author Response

Response to reviewer 2

I think its a very complete study, with an extensive literature review and with an easy to understand language for any reader, however, his contribution to new scientific knowledge is low. 

I believe that there are failures in the statistical methodology applied, because the observations of the response variables, are measured in the same animal during two conditions (normal and heat stress) which generates covariance between observations, which would have been resigned using a mixed effects model, which considers the types of variance covariance matrix thas is generated, which is not achieved with SAS PROC GLM, also allows the analysis of the breed x stress condition interaction, which is important to results interpretation.

Response: Thank you for comment and we agreed your suggestion. We performed statistical analysis as followed by your suggestion. We used two-way repeated measure ANOVA for physical heat-stress parameters (Figure 2). We also used mixed-effects model for biochemistry analysis (Figure 3-5) and flow cytometry results (Figure 6-8). So, some statistical significances were changed and figures were revised by new statistical methodology. However, our major finding did not change. We corrected figures, figure legends, and results during revision. We described statistical methods in detail in Material and Method section as well (2.6 Statistical analysis) (Line 188-198). In addition, we have attached the P-value of the effects of breed, environments, and breed x environments interaction on the statistical method used in the revised manuscript.

Reviewer 3 Report

In general the experiment has been conducted carefully and the results are presented clearly. There are however a couple of major issues.

1. The first is acknowledged by the authors in the final paragraph of the discussion. The normal readings were performed in May and the high THI in August, ie 3 months later. Cows were therefore at a later stage of lactation. This is a serious concern as many of the metabolic and immunological parameters measured will change with stage of lactation. It is not possible to compensate for this in the analysis. More information on the Days in milk at each time point must  be given.

2. Secondly, I have concerns over the feeding regime, see Lines 107-108. I am confused by the statement that feed was offered ad libitum but there were no orts. Firstly if no feed was left then it cannot have been ad libitum. Secondly, there  is much evidence as discussed in the paper that a high THI reduces feed intake, leading to many of the metabolic changes recorded. It is also stated that each cow was offered 20 kg of the same ration. The Holsteins had a greater body weight by about 180 kg and were shown to be yielding more milk by around 6-8 kg/d, so their nutrient requirements would be significantly greater. There would also be a change in yield over time associated with stage of lactation, so one would expect the requirements to decrease between the two time points analysed regardless of the environmental conditions.

3. The summary of which blood parameters were measured should be moved from the Results to the Methods section ie from Line 208 to Line 141. This also requires reference(s)  to the methodologies used. The name of the company which performed the work is insufficient.

4. For comparisons of immune cell populations between the breeds, some data regarding their health is required. For example did any cows used in the study have a raised somatic cell count? I am assuming none were clinically ill, but perhaps this should be stated.

5. It would have been useful to monitor body condition score changes over time. This would have supported the reasons suggested for some of the metabolite changes.

6. Lines 23, 37, 391. The two breeds of dairy cow are not different species. Please correct.

7. Lines 47-48. The description of heat stress seems back to front. It is more usually described as a form of hyperthermia in which the physiological systems of the body fail to regulate the body temperature within a normal range. This is because there is a greater input of heat from the environment together with internal heat production than the cow is able to dissipate through its control mechanisms.

8. L92. Cows are not laboratory animals.

9. L288 increased rectal temperature

10. Figs 6,7, 8. Please explain what the axis labels and different quadrants (Figs 6 & 8) or boxes (Fig 7) represent in the legend.

11. Formatting of references is inconsistent. Check punctuation and use of capital letters in journal titles.

12. The use of English is reasonable but sometimes incorrect so does need checking.

Author Response

Response to reviewer 3

In general the experiment has been conducted carefully and the results are presented clearly. There are however a couple of major issues.

  1. The first is acknowledged by the authors in the final paragraph of the discussion. The normal readings were performed in May and the high THI in August, ie 3 months later. Cows were therefore at a later stage of lactation. This is a serious concern as many of the metabolic and immunological parameters measured will change with stage of lactation. It is not possible to compensate for this in the analysis. More information on the Days in milk at each time point must be given.

Response: Thank you for valuable comment. First, we totally agree that metabolic and immunological parameters can be affected by confounding factors such as stage of lactation, parity number, or aging. To minimize these factors affecting the experimental results, we used Holstein and Jersey cows that similar DIM, parity number, and age of cows. we added more information and discussion on possible confounding factors of dairy cows in revised manuscript. (Line 97-109; 463-465)

Second, our study was conducted during the mid-lactation period, there was a three-month interval (May to August). Thus, we cannot fully exclude the effects of various factors for analysis in this experimental design. However, we used same Holstein and Jersey cows and it can be presented natural changes of metabolism and immune cell population of two breeds of dairy cow in normal to heat stress conditions. Collectively, we described limitation of our study and reinforced them in discussion part.

  1. Secondly, I have concerns over the feeding regime, see Lines 107-108. I am confused by the statement that feed was offered ad libitum but there were no orts. Firstly, if no feed was left then it cannot have been ad libitum. Secondly, there is much evidence as discussed in the paper that a high THI reduces feed intake, leading to many of the metabolic changes recorded. It is also stated that each cow was offered 20 kg of the same ration. The Holsteins had a greater body weight by about 180 kg and were shown to be yielding more milk by around 6-8 kg/d, so their nutrient requirements would be significantly greater. There would also be a change in yield over time associated with stage of lactation, so one would expect the requirements to decrease between the two time points analyzed regardless of the environmental conditions.

Response: Thank you for bringing up this point. First of all, there was a mistake in the expression about feeding regime. We have described better for this information in revised manuscript. (Line 113-118) Secondly, we agree that Holsteins be required more nutrients. However, the most important aspect of our study is the immune response and its physiological changes caused by heat stress. As you mentioned, feed intake is influenced directly under the heat stress. These results led us to reduce the effect on feed intake as much as possible in this study, and adopted a method to provide an adequate amount of ration (20 kg) to each cow.

  1. The summary of which blood parameters were measured should be moved from the Results to the Methods section ie from Line 208 to Line 141. This also requires reference(s) to the methodologies used. The name of the company which performed the work is insufficient.

Response: We have added references for metabolic assessment in revised manuscript (Line 166-170)

  1. For comparisons of immune cell populations between the breeds, some data regarding their health is required. For example, did any cows used in the study have a raised somatic cell count? I am assuming none were clinically ill, but perhaps this should be stated.
  2. It would have been useful to monitor body condition score changes over time. This would have supported the reasons suggested for some of the metabolite changes.

Response: All cows did not show any serious health issue during experimental period. We have added somatic cell count in milk for general health condition in M&M. Unfortunately, we could not obtain body scoring from this study. Future advanced research will take this into account. (Line 102-107)

  1. Lines 23, 37, 391. The two breeds of dairy cow are not different species. Please correct.

Response: Sorry for mistake. We corrected errors. (Line 23, 36, 423)

  1. Lines 47-48. The description of heat stress seems back to front. It is more usually described as a form of hyperthermia in which the physiological systems of the body fail to regulate the body temperature within a normal range. This is because there is a greater input of heat from the environment together with internal heat production than the cow is able to dissipate through its control mechanisms.

Response: Sorry for confusing. We newly described the sentence to make it clearer in revised manuscript. (Line 47-49)

  1. L92. Cows are not laboratory animals.

Response: Thank you for comment. We revised the word “laboratory animal” to “dairy cow” for clarity. (Line 93)

  1. L288 increased rectal temperature.

Response: Sorry for our mistake. We corrected error. (Line 318)

  1. Figs 6,7, 8. Please explain what the axis labels and different quadrants (Figs 6 & 8) or boxes (Fig 7) represent in the legend.

Response: Thank you for suggestion. We added information for each axis and gating strategy.

  1. Formatting of references is inconsistent. Check punctuation and use of capital letters in journal titles.
  2. The use of English is reasonable but sometimes incorrect so does need checking.

Response 11 and 12: Apologize for our mistake. We checked all of reference and corrected in conformity with Journal style in revised manuscript. Additionally, we corrected the inaccurate English for clear communication in revised manuscript.

Round 2

Reviewer 1 Report

The authors have appropriately addressed my previous concerns.

Author Response

Thanks for your efforts.  Thanks to your valuable comments, our manuscript now become better in shape.

Reviewer 2 Report

I believe that with the changes made to the article, suggested by the arbitrators, the documents has been substantially improved and its results are clearly presented

Author Response

(The authors gave the same response as above.)

Reviewer 3 Report

The authors have addressed the points raised regarding details of methods and better explanations of the figures. What they have been unable to achieve in this revised version is to correct for the two fundamental problems over the experimental design. The first is that it is impossible to distinguish between the changes due to the environment with those associated with a three month difference in the stage of lactation.  The second is that they fed both breeds of cows the same amount nutritionally, and repeat again that there were no orts but that the Holsteins were producing more milk. From this information it follows that the Holstein cows in particular were almost certainly not receiving as much food as they would have liked. This in turn will have affected their metabolite measurements and potentially their immune responses. I agree with the authors that the differences they report are more likely to have been due to the environmental changes than to either of these issues, but they cannot know this for certain. Some mention of the first problem is given in the final paragraph of the discussion. The second is not discussed at all. They really need to be more honest with their readers about this and address these concerns at the outset rather than somewhat half heartedly at the very end.

Other points.

I still have issue with the revised sentence at the start of the Introduction, lines 47-49. Body condition is a well used term in cows which relates mainly to body fat distribution. It is not the correct term to use here.

As well as the Table of dietary constituents, the ME and metabolizable protein contents of the diet need to be shown. A value of 10% CP seems very low. Calculations showing the extent to which the restricted diet provided met the metabolisable energy and metabolisable protein requirements of the cows according to their milk production are needed.

L213 should be consecutive not constitutive.

I don’t understand why the significance levels in the Tables are written as 0.05 < # p < 0.1. It should just be # p < 0.1.

Author Response

The authors have addressed the points raised regarding details of methods and better explanations of the figures. What they have been unable to achieve in this revised version is to correct for the two fundamental problems over the experimental design.

The first is that it is impossible to distinguish between the changes due to the environment with those associated with a three month difference in the stage of lactation.  

The second is that they fed both breeds of cows the same amount nutritionally, and repeat again that there were no orts but that the Holsteins were producing more milk.

From this information it follows that the Holstein cows in particular were almost certainly not receiving as much food as they would have liked. This in turn will have affected their metabolite measurements and potentially their immune responses. I agree with the authors that the differences they report are more likely to have been due to the environmental changes than to either of these issues, but they cannot know this for certain. Some mention of the first problem is given in the final paragraph of the discussion. The second is not discussed at all. They really need to be more honest with their readers about this and address these concerns at the outset rather than somewhat half heartedly at the very end.

Response: Thank you very much for your valuable comments. We agree that those issue could be the limitations or essential information of our study thus, we further improved an original manuscript during second round of revision. In revision, we added more discussion on these issues in the discussion part. We added 1) more discussion for potential confounding factors in this study. 2) discussion for limited amount of feeding for Holstein and Jersey cows. Please find revised manuscript (Line 465-482). A revised manuscript now conveys clearly potential issues in this study for readers.

Other points.

I still have issue with the revised sentence at the start of the Introduction, lines 47-49. Body condition is a well used term in cows which relates mainly to body fat distribution. It is not the correct term to use here.

Response: We deleted the sentence and add new sentence with reference for start of the introduction. The sentence includes negative impacts of heat stress in livestock. (Line 47-49)

Belhadj Slimen, I.; Najar, T.; Ghram, A.; Abdrrabba, M. Heat stress effects on livestock: Molecular, cellular and metabolic aspects, a review. J. Anim. Physiol. Anim. Nutr. 2016, 100, 401-412.

As well as the Table of dietary constituents, the ME and metabolizable protein contents of the diet need to be shown. A value of 10% CP seems very low. Calculations showing the extent to which the restricted diet provided met the metabolisable energy and metabolisable protein requirements of the cows according to their milk production are needed.

Response: Thanks for valuable comment. First of all, we added total digestible nutrients (TDN) of experimental diet, digestible energy (DE) and metabolizable energy (ME) in the revised manuscript as followed by your suggestion. (Line 117-120) We calculated DE and ME by applying the TDN energy conversion formula suggested by NRC (2001). The calculation formulas are as follows:

DE (Mcal/kg) = 0.04409 × TDN(%)

ME (Mcal/kg) = (DE × 1.01) – 0.45

Our experimental diet is higher than the required level of mid-lactation cows (milk yield 30kg, milk fat 4.3%, milk protein 3.4%, ME 52.0 Mcal/day) of Korean Feeding Standard for dairy cattle (2017).

For a value of CP, we realized that CP, NDF, and ADF data had wrong information in the chemical composition. We apologize for this big mistake. We corrected those values in the revised manuscript.

Rural Development Administration. 2017. Korean feeding standard for dairy cattle. National Livestock Research Institute, Republic of Korea.

National Research Council. 2001. Nutrient Requirements of Dairy Cattle: Seventh Revised Edition, 2001. Washington, DC: The National Academies Press. https://doi.org/10.17226/9825.

L213 should be consecutive not constitutive.

Response: Thank you for correction. We revised words to “consecutive”. (Line 139; 215)

I don’t understand why the significance levels in the Tables are written as 0.05 < # p < 0.1. It should just be # p < 0.1.

Response: Thank you for suggestion. We revised significance levels more simply in the figure legend and revised M&M section. (Line 199)
